# Capacitive neural network with neuro-transistors

Zhongrui Wang[1], Mingyi Rao[1], Jin-Woo Han[2], Jiaming Zhang[3], Peng Lin[1], Yunning Li[1], Can Li [1], Wenhao Song[1], Shiva Asapu[1], Rivu Midya[1], Ye Zhuo [1], Hao Jiang[1], Jung Ho Yoon[1], Navnidhi Kumar Upadhyay[1], Saumil Joshi[1], Miao Hu [3], John Paul Strachan[3], Mark Barnell[4], Qing Wu[4], Huaqiang Wu[5], Qinru Qiu[6], R. Stanley Williams [3], Qiangfei Xia [1] & J. Joshua Yang [1]

Experimental demonstration of resistive neural networks has been the recent focus of hardware implementation of neuromorphic computing. Capacitive neural networks, which call for novel building blocks, provide an alternative physical embodiment of neural networks featuring a lower static power and a better emulation of neural functionalities. Here, we develop neuro-transistors by integrating dynamic pseudo-memcapacitors as the gates of transistors to produce electronic analogs of the soma and axon of a neuron, with "leaky integrate-and-fire" dynamics augmented by a signal gain on the output. Paired with non-volatile pseudo-memcapacitive synapses, a Hebbian-like learning mechanism is implemented in a capacitive switching network, leading to the observed associative learning. A prototypical fully integrated capacitive neural network is built and used to classify inputs of signals.

[1] Department of Electrical and Computer Engineering, University of Massachusetts, Amherst, MA 01003, USA. [2] Center for Nanotechnology, NASA Ames Research Center, Moffett Field, CA 94035, USA. [3] Hewlett-Packard Laboratories, Palo Alto, CA 94304, USA. [4] Air Force Research Lab, Information Directorate, Rome, NY 13441, USA. [5] Institute of Microelectronics, Tsinghua University, Beijing 100084, China. [6] Department of Electrical Engineering and Computer Science, Syracuse University, Syracuse, NY 13244, USA. These authors contributed equally: Zhongrui Wang, Mingyi Rao. Correspondence and requests for materials should be addressed to R.S.W. (email: stan.williams@hpe.com) or to Q.X. (email: qxia@umass.edu) or to J.J.Y. (email: jjyang@umass.edu)

The computing archetype of the brain is not limited by the separation of memory and processing, serial execution, power inefficiency, and programming intensive issues of the von Neumann architecture[1–3]. Emerging devices[4–6] are a potential route to realize functionalities of neurons and synapses more efficiently than traditional complementary metal-oxide-semiconductor (CMOS) circuits and thus provide more capable neuromorphic systems. Memristors have recently demonstrated the integrate-and-fire capability of a McCulloch–Pitts model neuron[7–13]. Together with memristive synapses[5,14–29], fully memristive neural networks have been built with synapse–neuron interactions based on resistive coupling[30,31]. However, the signal gain accessible in a purely memristive circuit is limited. Natural neurons are remarkable for their ability to transmit action potentials, or spikes, over long distances and still drive an extremely large fan-out to communicate with hundreds of other neurons. All CMOS-based neuromorphic circuits use transistors to drive signals relatively long distances, but they lack the critical dynamics of biological systems. Thus, a hybrid approach is desirable, in which emerging memory devices provide the dynamics for neuromorphic functions and transistors supply the signal amplification to enable larger and multi-layer networks[32,33]. Moreover, such hybrid devices could exploit alternative mechanisms, such as capacitive rather than resistive coupling, to interact with its synapses, featuring a low static power dissipation to trigger active neuron operations[34–36].

We recently exploited the time-dependent electrical conductance resulting from the Ag mass migration due to the combined electrochemical[37–40] and diffusion processes[41] in a dielectric medium between two electrodes responding to an applied voltage to emulate the dynamics of ion channels in neurons[31,41]. Here in this work, we physically integrated such a diffusive memristor, which already possesses an intrinsic parallel capacitance, with a series capacitor to yield a memcapacitor-like circuit element that can significantly lower the power dissipation of the circuit because the signal is expressed as a voltage rather than a current. We utilize the resulting new element as the gate of a transistor, which then becomes the active front end of an "axon" for a stochastic leaky integrate-and-fire neuron emulator. Input signals are integrated by the new element that mimics dendritic spatial and temporal summation. A Hebbian-like mechanism was implemented to program pseudo-memcapacitive synapses with this hybrid neuro-transistor, which was then used for associative learning and signal classification in a prototypical integrated capacitive switching neural network.

## Results

### Temporal signal summation of dynamic pseudo-memcapacitors.
An ideal memcapacitor[36,42,43], if one existed, would possess a bias-history-dependent capacitance, similar to a memristor with resistance determined by the past inputs[44,45]. Different realizations of switching capacitors have been proposed[46–56]. For instance, electrochemical capacitors[56], bias-dependent polarization[46,48], and nanobattery effect[39,57] may all have the potential to implement memcapacitive systems. We developed a unique dynamic pseudo-memcapacitor (DPM) by integrating memristor and capacitor structures, as shown in Fig. 1a. The Pt/Ag/SiO$_x$:Ag/Ag/Pt diffusive memristor sits on top of a Pt/Ta$_2$O$_5$/TaO$_x$/Pt capacitor, which can also work as a one-selector-one-memristor cell once the bottom capacitor is electroformed to serve as a nonvolatile memristor at high voltage[58], providing this structure with multiple uses. Since Ta$_2$O$_5$ is a high-κ dielectric, the capacitance of the series capacitor $C_S$ is much larger than the intrinsic parallel capacitance $C_P$ of the diffusive memristor, which yields volatile unipolar capacitive switching similar to a memcapacitor (for simplicity, the memcapacitor circuit symbol is used in figures). Hysteresis loops of the logarithm of the charge vs. voltage were plotted for both biasing polarities (See Fig. 1b and Supplementary Figure 1) with the transition from capacitance state $C_P$ to $C_S$ occurring above an apparent voltage threshold. The applied electric field switched the diffusive memristor[41] after a delay to its low resistance or "ON" state, thus shorting out its intrinsic parallel capacitance. As a result, the overall capacitance of the integrated device, which was originally dominated by $C_P$, switched to $C_S$ (see Supplementary Figure 1 for more information on the impedance of diffusive memristor and DPM). The physical origin of the diffusive memristor threshold switching has been investigated and explained by its innate electro-thermal and ion migration dynamics[37,41,59]. A non-volatile memristor can replace the diffusive memristor in the structure shown in Fig. 1a to form a non-volatile pseudo-memcapacitor (NPM), which serves as the synapse of a fully capacitive neural network discussed later. Compared to memristors, DPM features low power consumption. As DPMs store energy in the electrostatic field rather than converting electricity to heat, the DPM-based capacitive networks feature low energy operation compared to networks built on other emerging devices (e.g., memristors) and are free of the sneak path leakage issue. The electrostatic energy is proportional to the capacitances of the elements. It shall also be noted, charging DPMs or capacitive elements takes energy away from the signal sources while discharging DPMs or capacitive elements returns the electrostatic energy back to the signal sources, as they essentially provide temporary energy storage. At steady states where signal sources output constant voltages, the power dissipation is theoretically zero. This clearly contrasts with the resistive neural networks, in which any non-zero signal will lead to Joule heating on resistive elements. In addition, the required energy for a single DPM to perform neural functions could be reduced by scaling the photolithography patterns, which also benefits the integration density (see Supplementary Note 1). Temporal signal summation is one of the fundamental functions performed by a single neuron, powered by the electrochemical gradients[39,57], which has been mathematically modeled by the Hodgkin–Huxley model[60]. The summation of signal in time is associated with the switching of the voltage-gated sodium and potassium ion channels, which integrate the post-synaptic potentials and initiate the subsequent action potential. As shown in Fig. 1c, in a typical temporal summation process, high-frequency pre-synaptic spikes propagate to the soma, which leads to the swift opening and shutting of a small portion of the sodium ion channels and the gradual stepping up of the membrane potential at time $t_1$. Once the membrane potential exceeds the threshold at time $t_2$, the fast inward-flow of sodium ions results in a significant further rise of the membrane potential. This positive feedback raises the potential rapidly until all available sodium ion channels are open, leading to the observed large upswing of the membrane potential in Fig. 1c. Once reaching the maximum, the membrane experiences repolarization at time $t_3$ due to the inactivation of the sodium ion channels and the opening of the potassium ion channels. By virtue of the biomimetic Ag dynamics[41], the response of a membrane potential to an input pulse train was replicated by the DPM. As shown in Fig. 1d, the DPM accumulated charge without "firing" when the potential across the series capacitance $C_S$ was low due to the non-linear OFF state resistance of the diffusive memristor at time $t_1'$. Such a non-linear $I–V$ relation, essentially an exponential function, makes the resistance small at a large voltage (easy to charge the capacitor) but large at a small voltage (slow to leak charge), mimicking the function of sodium ion channels. As the charge over the series capacitor increased, the voltage across the series capacitor rose with each subsequent pulse, closely reflecting the

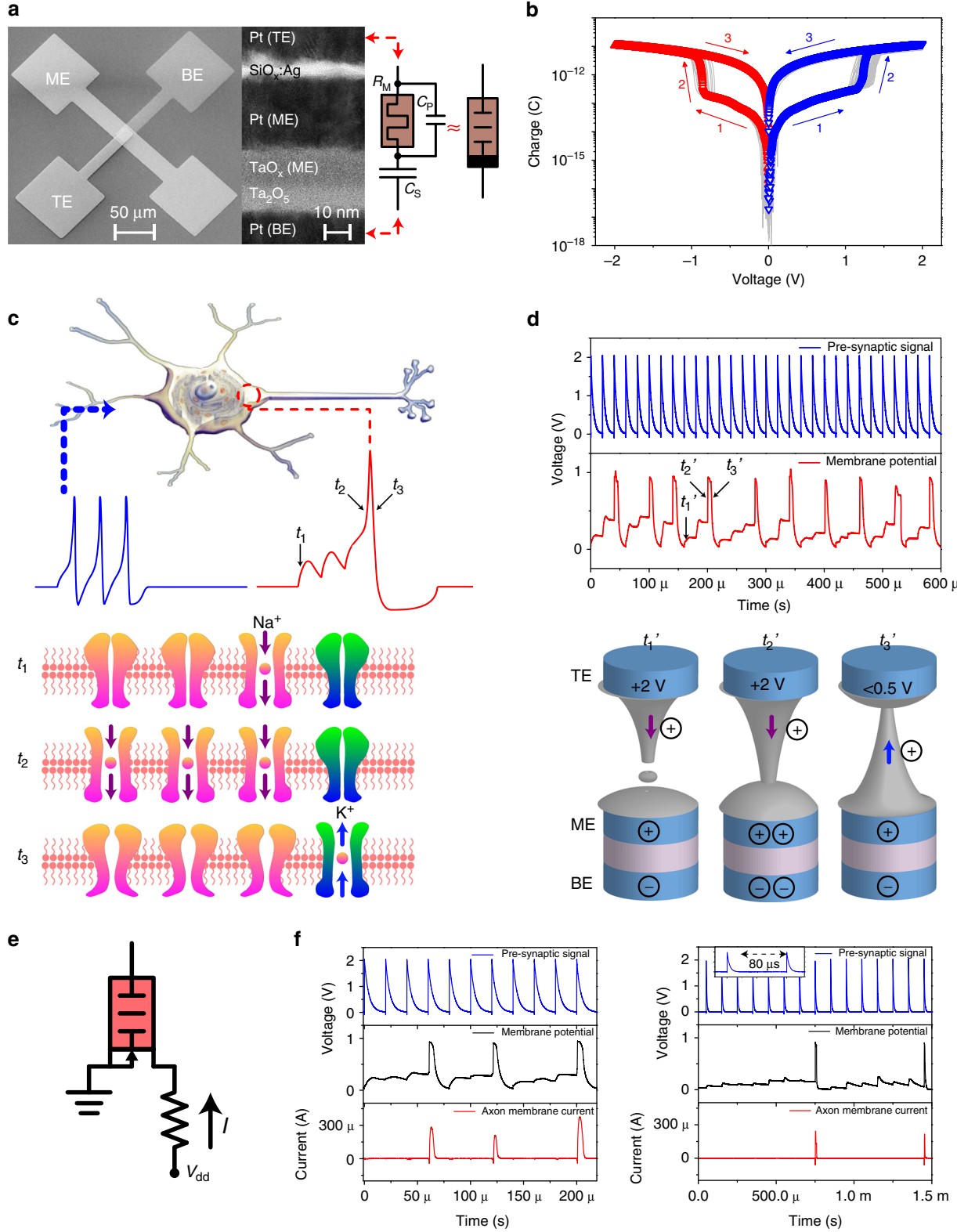

expected behavior illustrated for a neuron at $t_1$ in Fig. 1c. At time $t_2'$, the diffusive memristor was switched ON by the pre-synaptic spike to fully charge the capacitor, which replicated the upswing of the membrane potential due to the opening of all sodium ion channels at $t_2$ of Fig. 1c. At time $t_3'$, the pre-synaptic input was low. The memristor was first switched OFF and then switched ON again with opposite bias by the charged capacitor (see

Supplementary Figure 2 for the biasing dependent relaxation of the diffusive memristor), which quickly drained the capacitor and brought the membrane potential back to its resting value, similar to the repolarization caused by opening of potassium ion channels. The next pre-synaptic input spike would switch OFF the diffusive memristor, and start a new cycle of integrate-and-fire process. The volatility of the diffusive memristor essentially

**Fig. 1** Integrate-and-fire dynamics of the dynamic pseudo-memcapacitor (DPM). **a** Scanning electron micrograph of the plan view of the integrated DPM, and a transmission electron micrograph of the cross-section. **b** Charge–voltage relationship of the integrated DPM. The hysteresis loops reveal the volatile switching between the intrinsic parallel capacitor $C_P$ of the memristor and the series capacitor $C_S$. **c** Schematic representation of a biological neuron generating an action potential after receiving high-frequency post-synaptic inputs. At time $t_1$, the membrane potential did not reach the threshold so few sodium channels were open upon the arrival of the signal. As the neuron received further input stimuli, the membrane potential hit the threshold at time $t_2$ inducing quick opening of all available sodium ion channels. The potential reached its maximum and started to decrease due to the repolarization caused by the opening of the potassium ion channel and inactivation of sodium ion channels. **d** The integrate-and-fire process of a DPM. At time $t_1'$, the potential across the capacitor rose upon the input stimulus due to the nonlinear OFF state resistance of the diffusive memristor. At time $t_2'$, the diffusive memristor was switched ON by the pre-synaptic spike to fully charge the capacitor, which replicated the upswing of membrane potential due to the opening of all sodium ion channels at $t_2$ of **c**. At time $t_3'$, the pre-synaptic input was low. The memristor was first switched OFF and then switched ON again with opposite bias, which quickly discharged the capacitor and brought the membrane potential back to its resting value, similar to the repolarization caused by the opening of potassium ion channels. **e** Schematic of the neuro-transistor, consisting of a DPM integrated onto the gate of a MOSFET. **f** Dynamics of the neuro-transistor integrate-and-fire process. The left panel shows a sequence of input spikes at high frequency (20 μs intervals, blue line), the potential of the series capacitor (black lines), and the axon membrane current (red line) indicated in **e**. In the right panel, identical pulses with 80 μs intervals were applied, leading to a much slower firing rate

provides the repolarization and self-inactivation features of ion channels in neurons, which differs from non-volatile memristive neurons that require RESET pulses[8]. Compared to volatile memristive neurons with parallel capacitance[7,9,31], the non-polar diffusive memristor of the DPM plays the role of sodium ion channels in the phase of integration and the role of potassium channels to drain the stored charge and recover the potential in the phase of repolarization. Compared to single memristive neurons with intrinsic analog switching[8,11,31], DPMs provide physical embodiments of the cell membrane and ion channel, leading to the observed fidelity of neural function emulation.

The temporal integrate-and-fire behavior of the DPM enables the construction of an active neuron emulator by utilizing the switchable capacitance as the gate of a transistor, which then provides the amplification to drive the output down a signal line. The gate voltage mimics the membrane potential at the soma and the drain current replicates the electric charge spike or action potential flowing down the axon of a neuron (see Fig. 1e). A train of input pulses is integrated on the neuro-transistor gate to create a voltage spike as shown in Fig. 1f. The integration time and the duration of the firing events display statistical fluctuations because of the Ag migration dynamics in the diffusive memristors[61] (see Supplementary Figure 3). When the interval between input voltage spikes was increased, the volatile filament growth of the diffusive memristor meant that more spikes were required to build up the charge on the transistor gate to fire an action potential.

**Spatial signal summation and propagation of neuro-transistors**. The dendritic tree of each biological neuron interfaces with adjacent branched axon projections via synapses to propagate the electrochemical stimulation from neighboring neurons to its own soma[60]. The morphology of dendrites varies, providing neurons with different functionalities[60]. The temporal nature of signals fed to the soma plays a fundamental role in summing synaptic inputs and in determining whether an action potential is produced. The spatial summation of synaptic inputs is highly nonlinear, which enhances information processing capabilities at the single neuron level[33]. The DPM gate of the neuro-transistor calculates the temporally weighted sum of the input signals and thus modulates the corresponding drain output current based on the result, in a fashion similar to a biological neuron[32].

Figure 2a–d shows circuits with two capacitors, $S_1$ and $S_2$, that act as synapses connected to the DPM gate of a neuro-transistor. The red and blue pulse trains of Fig. 2a, c represent the input signals from neighboring neurons. The triggered gate potential and output current are depicted by the black and green curves,

respectively, in Fig. 2b, d. In the case of Fig. 2a, b, only a single synapse was excited and the resultant voltage stimulation was not sufficient to fire the neuro-transistor. However, concurrent stimulation of both synapses, shown in Fig. 2c, d, produced clear firing as revealed by the gate potential and the output current spikes in Fig. 2d. This spatial summation results in super-linearity of the firing rate at the gate because the delay of a diffusive memristor decreases exponentially with the voltage[41,61]. Since the two input voltage signals were added in the form of charge accumulation on capacitors, ideally there was no static power dissipation as compared to the weighted sum of currents in a resistively coupled neural network[34].

Biological neurons are gain elements because ion pumps convert chemical energy to electrical potential, which makes neurons active devices. To achieve this function with an electronic emulator, the artificial neuron should be able to provide energy for signal fan-out and propagation in multi-layer networks[62,63]. Recent neuromorphic designs have utilized passive synapses[5,14–21], relying on active elements such as operational amplifiers to change synaptic weights and compute inferences from their aggregate states. The neuro-transistor introduced here features the minimal circuit footprint among all active neuron-like devices (with ~10 transistors or more) reported to date[3,9,64,65], with the DPM functionality integrated directly onto the gate of a single transistor. The transistor provides the signal propagation, for example in the two-stage circuit shown in Fig. 2e. The first stage neuro-transistor $N_1$ (based on an n-MOSFET) integrated several pulses from input signal 1 and fired (see Fig. 2f). The weighted sum of the output of $N_1$ expressed on the capacitive synapse $S_1$ and the input signal 2 led to the observed integrate-and-fire behavior of the second stage neuron $N_2$ (based on a p-MOSFET), as revealed by the current output shown in Fig. 2g, illustrating two important biomimetic properties: neuronal gain and spatial summation.

**Associative learning in pseudo-memcapacitive networks**. Hebb's rule is one of the most important cellular mechanisms for synaptic weight modulation, in which correlated pre- and post-synaptic signals modify the synaptic weight[66]. To realize a Hebbian-like mechanism in a capacitive neural network, the neuro-transistor was paired with NPM synapses, built using a similar structure but with non-volatile electrochemical metallization cells to replace the volatile diffusive memristors (see Methods and Supplementary Figure 4). Since the capacitance weight change was persistent, it simulated the long-term plasticity of chemical synapses such as potentiation and depression. The firing of the post-synaptic neuron was accompanied by an increase of the effective capacitance. A back-propagating signal

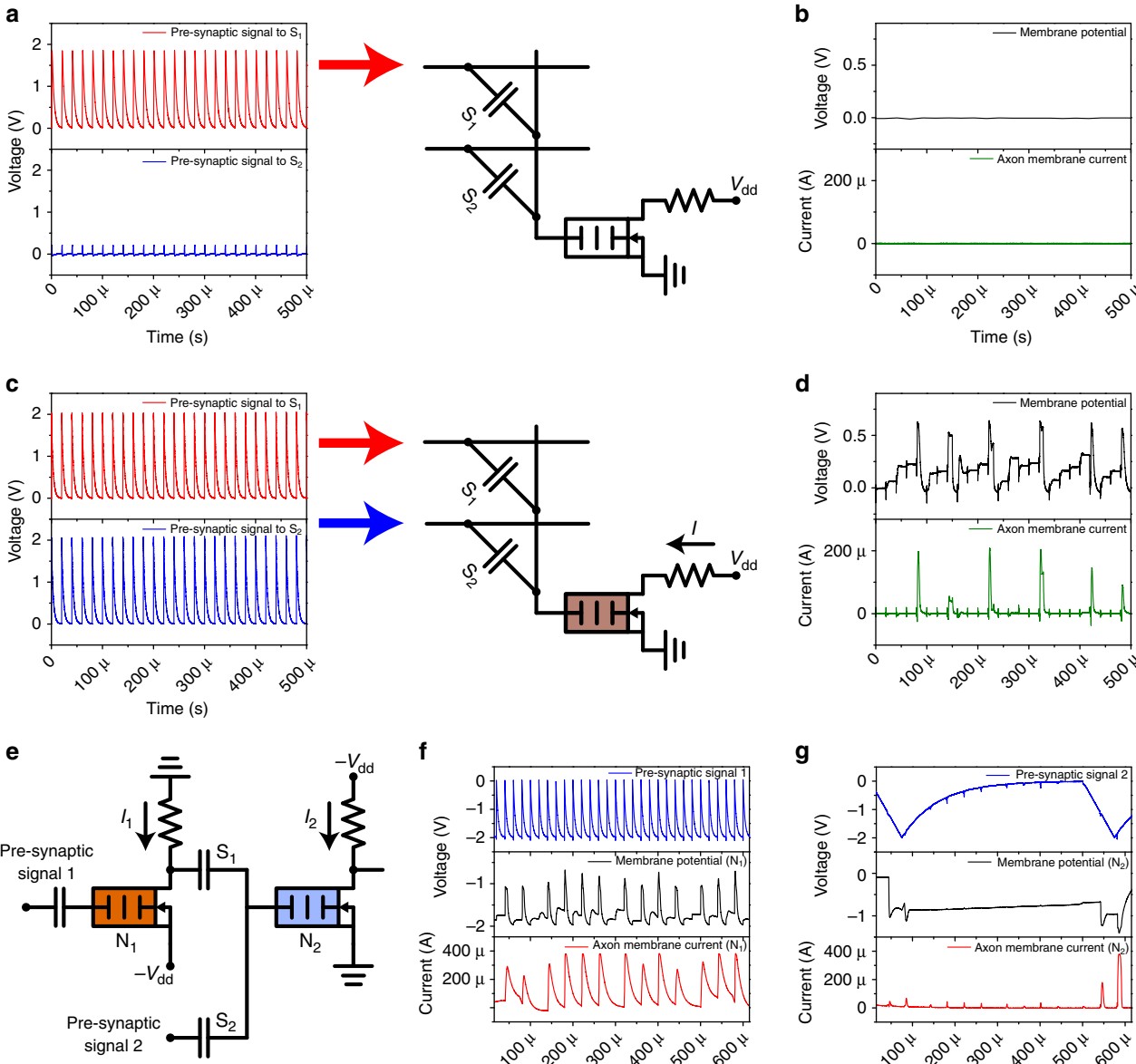

**Fig. 2** Dendritic spatial summation and signal propagation of neuro-transistors. **a**, **c** Input signals and circuit diagrams. The red and blue curves represent the "pre-synaptic" voltage spikes to two capacitive synapses. The DPM serves as the gate of the neuro-transistor with two capacitively-coupled fixed-weight synapses $S_1$ and $S_2$. **b**, **d** The triggered gate potential, representing the membrane potential of a biological neuron, and the drain current, corresponding to the axon current of the neuro-transistor, depicted by the black and green curves, respectively. In the case that a single synapse was excited, the resultant stimulation could not trigger output pulses. However, concurrent stimulation of both synapses produced clear temporal integration on the neuro-transistor gate and the associated spikes in the output current. **e** Circuit diagram of cascaded neuro-transistors $N_1$ (n-MOSFET) and $N_2$ (p-MOSFET). Fixed capacitance synapses were used with equal weights (680 pF). **f** The first stage $N_1$ integrated the fixed-frequency input signal 1 (blue line) and fired periodically (black and red lines), with some variation in the integration time. **g** The voltage to the second stage $N_2$ was the weighted sum of the voltage on $S_1$ resulting from the output of $N_1$ and the input signal 2 (blue line), with the resulting $N_2$ gate potential (black line) and the second stage output current (red line) shown. The firing of the second stage neuro-transistor $N_2$ illustrates both spatial summation and sustainable signal propagation

was transmitted to the synapse because of the voltage division between the capacitive front end of the neuro-transistor and the synapse capacitor, which could program the synapse to a high capacitance state (HCS) together with the firing of the pre-synaptic neuron. Such backpropagation, which usually requires complicated active feedback circuitry in conventional CMOS solutions, was implemented in a straightforward and reliable approach. Thus, the potentiation of the synapse temporally correlated forward-propagating signals from the pre-synaptic neuron with back-propagating signals from the DPM gate of the post-

synaptic neuron in an unsupervised fashion, forging the basis for the Hebbian-like learning (see Supplementary Figure 5). Utilizing this principle, we demonstrate associative learning with a capacitive neural network for the first time, which is implemented in a time-division multiplexing scheme (see Methods) here and had been previously demonstrated using non-capacitive elements including memristors to show fundamental learning processes like classical conditioning[4,67–69]. In classical conditioning, associative learning involves repeatedly pairing an unconditioned stimulus, which always triggers a reflexive response, with a

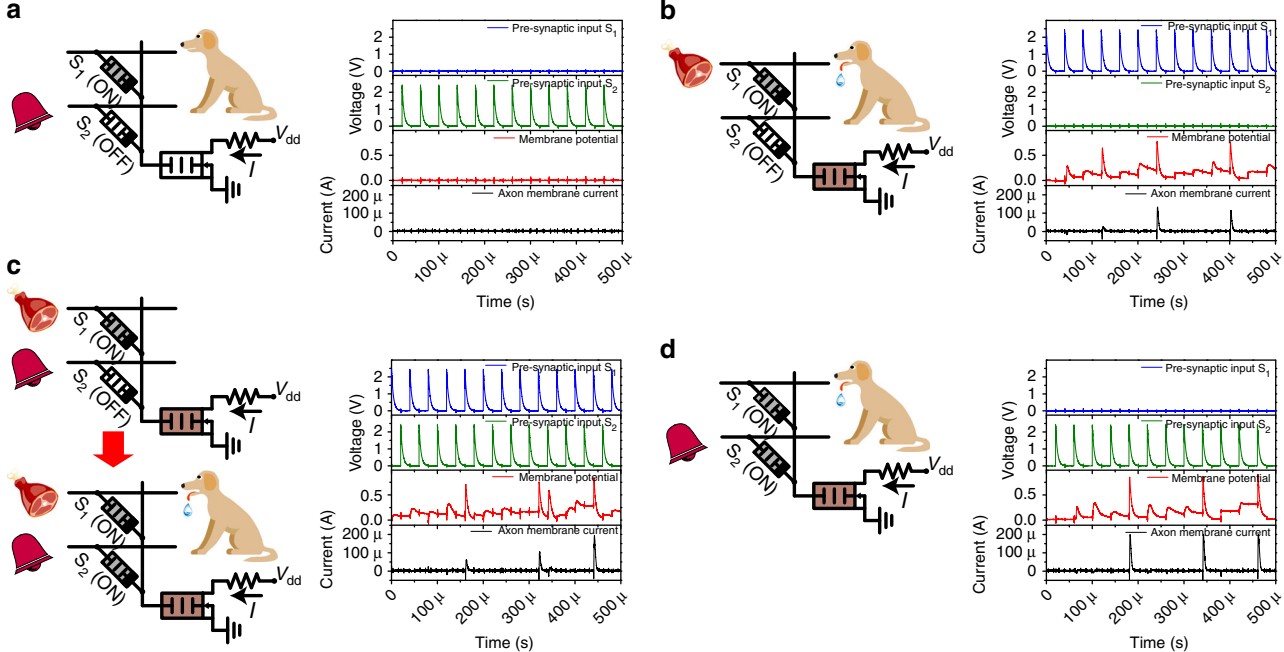

**Fig. 3** Capacitive network for associative learning based on the Hebbian-like mechanism. Two pre-synaptic signals model the sight of food and the sound of a bell, respectively. The post-synaptic neuron models the salivation of a dog. The initial weight of the synapse interfacing with the "food" pre-synaptic neuron was large, while that of the synapse connected to the "bell" pre-synaptic neuron was small. **a** Probing with only the "bell" signal before association. The post-synaptic neuron did not fire. **b** Probing with only the "food" signal triggered the firing of the post-synaptic neuron. **c** Associative learning with simultaneous "bell" and "food" stimuli applied for a sufficient time. If the relaxation time after firing of the "salivation" pre-synaptic neuron was long enough, the backpropagating signal could overlap with the forward propagating voltage spikes from the "bell" neuron, which programmed the non-volatile pseudo-memcapacitor (NPM)-based "bell" synapse. **d** Probing after the conditioning phase. The post-synaptic neuron fired with only the "bell" stimulus

neutral stimulus, which normally triggers no response. An association is developed between the neutral stimulus and the unconditioned stimulus after training, so that the response can be triggered by either the unconditioned stimulus or the neutral stimulus, with the result that the latter becomes a conditioned stimulus.

As shown in Fig. 3a–d, two pre-synaptic neurons, represented by voltage sources, modeled the sight of food and the sound of a bell, respectively. The post-synaptic neuron modeled the salivation of a dog. The initial weight of the synapse interfacing with the "food" pre-synaptic neuron was large, while that of the synapse connected to the "bell" pre-synaptic neuron was small. It should be noted that our associative neural network is symmetrical. The signals of the stimulus could be swapped with affiliated synaptic weights, which contrasts to asymmetric schemes for which inputs cannot be exchanged[68]. Figure 3a, b illustrates the "probing phase" before the association; individual food or bell stimuli were fed to the corresponding neurons. The firing of the "salivation" post-synaptic neuron was only triggered by the "food" signal but not that of the "bell" neuron, as the large capacitance weight of the "food" synapse resulted in a sufficient voltage drop across the "salivation" neuron to be integrated beyond the threshold (See Fig. 3b). No synaptic weight was changed in the probing stage. The process of association is depicted in Fig. 3c, for which simultaneous "food" and "bell" stimuli were applied for a sufficient time. The "salivation" post-synaptic neuron fired due to the stimulus from the "food" synapse. The DPM neuron showed a short and stochastic relaxation time because of its underlying dynamics (see Supplementary Figure 3). There was a probability that a relaxation process would last for more than one spike period (e.g., 20 µs in Fig. 3c). Once a long relaxation occurred, the firing

of the "salivation" neuron (now with a large gate capacitance) could overlap with the forward propagating voltage spikes from the "bell" neuron, yielding a sufficiently large voltage to potentiate the "bell" synapse and thus create the association between the "bell" and the "salivation". In Fig. 3d, the association was verified using only the "bell" stimulus, which successfully triggered the "salivation" neuron. The implementation of this classical conditioning verifies the Hebbian-like learning in a fully capacitive neural network (see Supplementary Figure 6 for the other scenarios of the Hebbian-like mechanism).

**Fully integrated pseudo-memcapacitive networks**. A prototype chip with a fully integrated capacitive neural network was fabricated with the architecture illustrated schematically in Fig. 4a. The NPM synapses formed a 4 × 4 crossbar array (blue box) with DPM neurons (red box) at the end of each column (see Fig. 4a, b), creating a fully connected network. Such a spiking neural network closely resembles the biological counterpart.

The neuro-transistors and synapses were constructed by vertically stacking diffusive memristors and electrochemical metallization cells, distinguished by the relative amount of Ag utilized in each, with a series capacitance, respectively (see Fig. 4b, c). The neuro-transistor consisted of a diffusive memristor integrated onto the gate of a conventional n-MOSFET (see Fig. 4e). Structural analysis using high-resolution cross-sectional TEM revealed that the $SiO_x$ dielectric matrix was amorphous and the thin Ag layer was nano-crystalline, as shown in Fig. 4f. The volatile resistive switching is related to the interfacial energy minimization or Gibbs–Thomson effect, which causes bridging Ag nanoparticles in the matrix to ripen and coalesce onto the electrodes[41,70] (see Supplementary Figure 7). The compositional

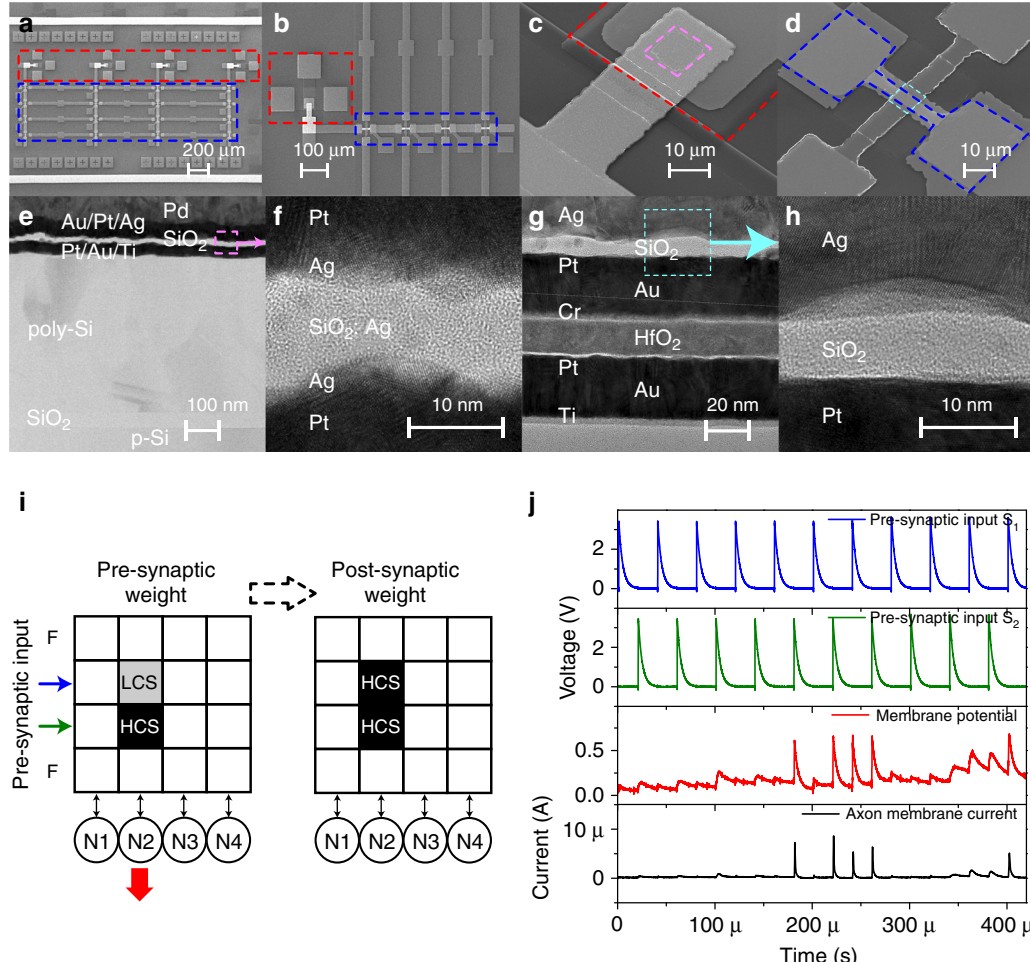

**Fig. 4** Prototypical fully integrated capacitive neural network. **a** The scanning electron micrographs of the network consisting of a 4 × 4 crossbar array of synapses (blue box) linked with four neurons (red box). **b** Each neuron was interfaced to four synapses. **c**, **d** The DPM neurons and the NPM synapses were both built utilizing Ag-based memristors (magenta and cyan boxes, respectively) in series with integrated capacitors (red and blue boxes). **e** Structural analysis of the neuro-transistor. The cross-sectional transmission electron micrograph shows the diffusive memristor sitting on the gate of a convention n-MOSFET. **f** The high-resolution transmission electron micrograph of the diffusive memristor showing the amorphous $SiO_x$ dielectric matrix with nano-crystalline thin Ag layers. **g** Structural analysis of the integrated synapse. The transmission electron micrograph shows the electrochemical metallization cell on top of a $HfO_2$ capacitor. **h** The zoomed high-resolution transmission electron micrograph of the electrochemical metallization cell shows the thick Ag electrode responsible for the longer retention time of the high conductance state of the memristor. **i**, **j** Schematic illustration and experimental observation of the synapse programming using the Hebbian-like mechanism. Simultaneous pre-synaptic signals (blue and green lines in **j**) were applied to both the low capacitance state (LCS) synapse and the high capacitance state (HCS) synapse. The post-synaptic neuron fired (red lines in **j**) and the LCS synapse was subsequently potentiated, which made the neuron fire within a shorter integration time from 280 to 400 μs (red curve of **j**)

information is further confirmed by the energy dispersive X-ray spectroscopy (EDS) data (see Supplementary Figure 8). On the other hand, each of the NPM synapses consisted of an electrochemical metallization cell on top of a series high-κ $HfO_2$ capacitor (see Fig. 4g and Supplementary Figure 8), which featured a thick Ag electrode and larger Ag co-sputtering power to increase the concentration of Ag in the dielectric matrix and thus a significantly longer retention time to meet the requirements of synapses (see Fig. 4h). The integrated capacitive neural network is compatible with existing infrastructures for silicon-based technologies and subjected to the same scaling capability of transistors. (The launch of extreme ultraviolet lithography could potentially make the gate cross-section a similar size to that of the filament of memristors[59].) The time-multiplexing pre-synaptic inputs were generated off-chip (see Methods).

As illustrated in Fig. 3, the programming of the synaptic array was realized by a Hebbian-like mechanism. In Fig. 4i, j, a low

capacitance state (LCS) synapse could be potentiated if both the pre-synaptic (blue arrow in Fig. 4i and blue lines in Fig. 4j) and post-synaptic neurons (red arrow in Fig. 4i and red lines in Fig. 4j) fired together. The firing of the post-synaptic neuron could be triggered by stimulating adjacent synapses in the HCS for a sufficiently long period (green arrow in Fig. 4i and green lines in Fig. 4j). The post-synaptic neuron fired because of the stimulus from the HCS synapse. Stochastic variations in the firing pattern of the post-synaptic neuron (red lines in Fig. 4j, also see Supplementary Figure 3) led to the spike overlap with the pre-synaptic input, which yields a sufficiently large voltage to SET the electrochemical metallization cell of the synapse, i.e., programs the synapse from LCS to HCS.

Pre-synaptic signals could be fed into and classified by a fully connected neural network because of the intrinsic vector–matrix multiplication capability, which has so far been demonstrated on resistive crossbar arrays only[22,28]. Here we show that capacitive

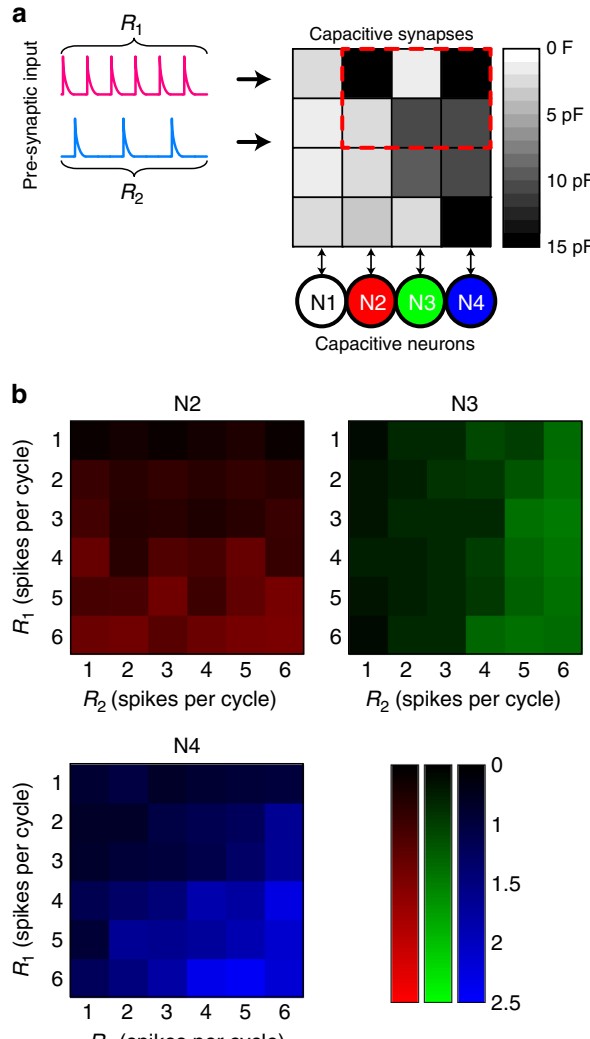

**Fig. 5** Inference of the pre-synaptic signals with the capacitive neural network. **a** Schematic of the classifier. A 2 × 3 subarray of the neural network, as highlighted by the red dashed box, was used to classify the two-dimensional input vector consisting of non-overlapping voltage spikes at different rates, $R_1$ and $R_2$ spikes per period (see also Supplementary Figure 10). The NPM synapses were initialized to discrete states. **b** Color maps of the post-synaptic neuron integrate-and-fire rate per input cycle (only the first integrate-and-fire cycle counts) as a function of the spiking rates of the two pre-synaptic inputs

crossbar in Fig. 4 also bears in-memory dot product ability without the need of digital logic, thus performing inference on pre-synaptic signals encoded by spiking rates. The synapses of the network were pre-programmed to different weights (see Supplementary Figure 9 for the measurement of the equivalent capacitance of synapses). A simple inference is demonstrated using a 2 × 3 subarray interfaced with three neurons and receiving a two-dimensional input represented by the rates (frequencies) of input spikes (see Supplementary Figure 10 for the input patterns per cycle). Inputs to the network were mapped to the output space represented by the integrate-and-fire rate (cycle duration divided by the integration time before the first fire of the neuron) of the post-synaptic neurons (e.g., N2, N3, and N4) as shown by the color maps in Fig. 5b. As the array might be programmed during this interference process, the array was re-programmed to the same pattern before each inference cycle. This proof of

principle demonstration for the spiking network experimentally verified the feasibility of using a capacitive neural network for signal classification[54].

## Discussion

We have demonstrated the capability of neuro-transistors based on a dynamic pseudo-memcapacitive gate to better emulate neural functions at reduced power consumption and footprint, which exhibits stochastic leaky integrate-and-fire. The active operation of the neuro-transistors enables sustainable signal propagation and spatial summation in capacitive artificial neural networks with passive synapses. Paired with non-volatile pseudo-memcapacitive synapses also developed in this study, a Hebbian-like learning mechanism was demonstrated, which naturally exhibited associative learning. Built on these newly developed neurons and synapses, the prototypical integrated capacitive neural network with classification capability has shown the promise as an alternative energy-efficient and bio-faithful routine for the hardware implementation of neuromorphic computing.

## Methods

**Fabrication of DPM**. A vertical stack for the diffusive memristor and capacitor was patterned by conventional photolithography on a p-type (100) Si wafer with 100 nm thermal oxide. The Pt bottom electrodes (BE) of 20 nm thickness were evaporated onto the SiO$_2$ substrate with a 2 nm thick Ti adhesion layer. A 10 nm thick blanket Ta$_2$O$_5$ dielectric layer was deposited by sputtering a Ta$_2$O$_5$ target in Ar and O$_2$ plasma as the dielectrics of the series capacitor. The middle electrode (ME) consists of a Ta-rich TaO$_x$ layer grown by evaporating 10 nm Ta covered by another 20 nm evaporated Pt. A 10 nm SiO$_x$ buffer layer was grown by sputtering a SiO$_2$ target in Ar plasma and then treated by reactive ion etching (mixed CHF$_3$ and O$_2$ gas plasma) to form contact holes to the MEs. The holes were then filled with a 2 nm sputtered Ag layer. A 10 nm thick SiO$_x$:Ag switching layer of the diffusive memristor was deposited by co-sputtering SiO$_2$ (RF power 270 W) and Ag (RF power 14 W) targets in Ar. Top electrodes (TE) of 2 nm Ag and 30 nm thick Pt were evaporated on the switching layer.

**Discrete active neuro-transistor**. The diffusive memristor was fabricated in a similar manner as the DPM by omitting the series capacitor. The neuro-transistor based on the discrete device was constructed by wiring the BE of the diffusive memristor to the gate of the requisite transistors (BSH103 and BSH203, Nexperia USA Inc.).

**Discrete electrochemical metallization cell**. Pt BEs of 20 nm thickness were evaporated onto the SiO$_2$ substrate with a 2 nm thick Ti adhesion layer. A 8 nm thick SiO$_x$:Ag switching layer was deposited by co-sputtering SiO$_2$ (RF power 270 W) and Ag (RF power 20 W) targets in Ar. TEs of 10 nm Ag and 30 nm thick Pt were evaporated on the switching layer.

**Fully integrated memcapacitive switch network**. A p-type (100) substrate was used as a starting material. A field oxide of approximately 500 nm depth is defined by local oxidation. A silicon dioxide of 30 nm was thermally grown as a gate dielectric, followed by in-situ doped n$^+$ polysilicon deposition. After patterning the polysilicon gate, the source and the drain are formed by phosphorous doping. The channel length and width of the fabricated transistor were 50 and 80 μm, respectively. The bottom metal of the diffusive memristor was patterned on the gate of the transistor, followed by CF$_4$ reactive ion etching to remove native oxide. A 2/4/15/5 nm Ti/Ag/Au/Pt electrode was evaporated on the naked gate. A 10 nm thick SiO$_2$ passivation layer was deposited to cover the gate. Contact holes to the BEs of the diffusive memristor were etched by CHF$_3$ and O$_2$ plasma which were filled with 2 nm Ag. The 10 nm thick SiO$_x$:Ag switching layer of the diffusive memristor was patterned and deposited by co-sputtering SiO$_2$ (RF power 270 W) and Ag (RF power 14 W) targets in Ar. TEs of 2 nm Ag and 30 nm thick Pt were patterned and evaporated. The NPM synapses were grown on the extended TE of the DPM neuron. An 8 nm HfO$_2$ ALD dielectric layer providing series capacitance was grown and patterned, followed by 2/20 nm Cr/Pt as the BEs for the Ag-based electrochemical metallization cells. The 8 nm thick SiO$_x$:Ag switching layer of the metallization cells was deposited by co-sputtering SiO$_2$ (RF power 270 W) and Ag (RF power 20 W) targets in Ar. TEs of 10 nm Ag and 100 nm thick Pd were patterned and sputtered.

**Electrical measurements**. Electrical measurements of the charge–voltage characteristics of the DPM was performed on a Keysight B1500A semiconductor device analyzer equipped with a B1530A waveform generator/fast measurement unit

(WGFMU). The triangular waveform with a 0.016 V $\mu s^{-1}$ slope was employed. The stored charge was calculated by integrating the sensed current over time.

Characterization of the active neuro-transistor was performed using the Keysight 33622A arbitrary waveform generator and the Keysight MSOX3104 mixed signal oscilloscope. The oscilloscope probes are of capacitance much smaller than that of the HCS of the DPM (e.g., gate capacitance of the transistor). Pre-synaptic signals were applied using the built-in waveforms of the Keysight 33622A. The analog oscilloscope channels were used to measure the voltage at the output of the function generator, drain of the transistor, and that across the DPM.

For the demonstration of the associative neural network in Fig. 3, synapse programming in Fig. 4, and signal classification in Fig. 5, a time-division multiplexing scheme was employed that the pre-synaptic signal to the upper synapse would be active in the first period (e.g., 20 µs) while the input to the lower synapse (always at zero potential in this period) would be high-impedance or floating (see Supplementary Figure 10). This was reversed in the next period of 20 µs. The input was only floating when its signal was at zero potential, which maps "0" to high-impedance equivalently. The high-impedance input was realized with series AD8180 multiplexers (Analog Devices Inc.) connecting the pre-synaptic signal outputs from the Keysight 33622A.

**Data availability**. The data that support the findings of this study are available from the corresponding author upon request.

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

## Acknowledgements

This work was supported in part by the U.S. Air Force Research Laboratory (AFRL) (Grant No. FA8750-15-2-0044), the Defense Advanced Research Projects Agency (DARPA) (Contract No. D17PC00304), and the National Science Foundation (NSF) (ECCS-1253073). Any opinions, findings, and conclusions or recommendations expressed in this material are those of the authors and do not necessarily reflect the views of AFRL. H.W. was supported by the Beijing Advanced Innovation Center for Future Chip (ICFC) and NSFC (61674089, 61674092). Part of the device fabrication was conducted in the clean room of the Center for Hierarchical Manufacturing (CHM), an NSF Nanoscale Science and Engineering Center (NSEC) located at the University of Massachusetts Amherst.

## Author contributions

J.J.Y. and Z.W. conceived the concept. J.J.Y., Q.X., R.S.W., Z.W. and M.R. designed the experiments. Z.W. and J.W.H. fabricated the devices. Z.W. and M.R. performed electrical measurements. J.Z., P.L., Y.L., C.L., W.S., S.A., R.M., Y.Z., H.J., J.H.Y., N.K.U., S.J., M.H., J.P.S., M.B., Q.W., H.W. and Q.Q. helped with experiments and data analysis. J.J.Y., Q.X., R.S.W., Z.W. and M.R. wrote the paper. All authors discussed the results and implications and commented on the manuscript at all stages.

## Additional information

**Competing interests:** The authors declare no competing interests.

