## [Peer Review File · Nature Communications]

Reviewers' comments:

Reviewer #1 (Remarks to the Author):

This paper presents an experimental realization of a memcapacitive neural network. The pseudo-memcapacitors are realized using memristors and capacitors by employing both intrinsic and extrinsic capacitances. The network is then trained to achieve associative learning (classical conditioning).

The paper is well written and clear. I find the work innovative in that it shows a practical realization of a memcapacitive neural network which could be useful in low power neuromorphic computing.

It would however be interesting (and important) if the authors would comment on the scalability of such networks in terms of power needed and integration. They present very small network sizes and yet they say that their work promises a "bio-plausible" alternative for "hardware implementation of neuromorphic computing".

It is true that their structures express signals in terms of voltages rather than currents, but how does the power scale with size? And, how difficult is it to integrate the proposed network for larger instances that ultimately will have to be realized if one is seeking practical applications?

Reviewer #2 (Remarks to the Author):

The manuscript by Wang et al. reports very interesting results on combining the functionalities of two different type devices based on Ag:SiO₂ and Ta₂O₅ implementing them in a neuro-transistor. The authors demonstrate a Hebbian-type (associative) learning mechanism. The experiments are carefully performed and the manuscript is well structured and written. The topic is suitable for the journal.

I would recommend the manuscript for publication, provided the authors address some issues on the interpretation of the mechanism of operation.

Comments:

1. The authors suggested that Pt/Ta₂O₅/TaO_x/Pt cell is a capacitor. However, several groups have reported that device with this structure works as a memristor (e.g. Nat Commun 2013, doi: 10.1038/ncomms3382). Could the authors provide an experimental proof that the cell is indeed memcapacitive and not memristive? This point is important to be justified as it influences the whole discussion and is also present in the title.

2. The cell is very similar to CRS cells, employing one ECM and one VCM cell. Is it possible that this combination is providing the functionalities and not a memcapacitive effect?

3. The authors are discussing in several paragraphs in the text that their circuit is bio-similar. However, in the living organisms the transmission of the signals occurs based on generated diffusion potential (being a part of the nanobattery effect). To my best knowledge there is no reported biological cell using (bio)capacitors. From the text it can be made a misleading conclusion that implementing capacitors makes the circuit more bio-similar.

4. It will be of benefit for the readers if it is more clearly explained what is the advantage of the present work/approach compared to other recent papers published in this field (by the authors and other groups). Most of the recent research in the field is claiming on bio-similarity and it is important to compare and explain in more detail.

Minor comment:

On page 2 the authors wrote "...conductance resulting from the diffusion of nanoscale Ag clusters in a dielectric medium..." I

The sentence is not correct. The Ag-clusters are not directly diffusing but dissolve or agglomerate or are displaced. The Ag⁺ ions are diffusing (more precisely – migrating in the high electric field).

Reviewer #1 (Remarks to the Author):

This paper presents an experimental realization of a memcapacitive neural network. The pseudo-memcapacitors are realized using memristors and capacitors by employing both intrinsic and extrinsic capacitances. The network is then trained to achieve associative learning (classical conditioning).

The paper is well written and clear. I find the work innovative in that it shows a practical realization of a memcapacitive neural network which could be useful in low power neuromorphic computing.

Reply: We thank the referee for the positive comments on the significance of this work.

It would however be interesting (and important) if the authors would comment on the scalability of such networks in terms of power needed and integration. They present very small network sizes and yet they say that their work promises a “bio-plausible” alternative for “hardware implementation of neuromorphic computing”. It is true that their structures express signals in terms of voltages rather than currents, but how does the power scale with size? And, how difficult is it to integrate the proposed network for larger instances that ultimately will have to be realized if one is seeking practical applications?

Reply: We thank the referee for this question. Theoretically, all elements of the pseudo-memcapacitive network chip could laterally scale with the same factor, which ideally does not introduce any change to the node voltages but reduce both chip area and operating power after scaling. In addition, the scaling of a single volatile pseudo-memcapacitor would not affect the integrate-and-fire behavior. Therefore, the scaled pseudo-memcapacitive network chip could reduce the operating power and be integrated like that of the microscale chip, as discussed in detail below. The discussion text is arranged as following: (1) Scaling impact on network integration and function (network node voltages and single pseudo-memcapacitor neuron integrate-&-fire behavior). (2) Scaling impact on the operating power.

(1) **Scaling impact on network integration and function.** Here we discuss the scaling of all lateral lithography mask patterns by a factor k while keeping vertical dimensions (e.g. deposition thickness) unchanged. Such lateral scaling could be easily implemented by varying the magnification of the reduction lens in mask stepper systems. It is also worth noting that the launch of 3nm EUV lithography process¹ could potentially make the neuro-transistor gate cross-section a similar size with that of a memristor filament², which indicates the potential of ultrahigh density integration. The capacitance of a basic building block, a Metal-Insulator-Metal (MIM) stack, after scaling is given by the equation below.

$$C_{after} = \epsilon \frac{A}{k^2 d} = \frac{1}{k^2} C_{before}$$

Note that C_{after} and C_{before} denote the capacitance after and before scaling, respectively. The dielectric permittivity is ϵ . The lateral area of the device before scaling is A . The dielectric thickness is d . Since every element is subjected to the same scaling factor, the capacitance ratio between arbitrary two elements remains the same after scaling. It shall be noted that the scaling also applies to parasitic capacitance (e.g. substrate parasitic capacitance). Therefore, the network node voltages of the chip after scaling all lithography patterns with the factor k should not be affected, which is also illustrated in Figure R1. Figure R1a shows the schematic of

the 4×4 network of Fig. 5 of the main text, consisting of 4 pre-synaptic neurons and 4 post-synaptic neuron capacitors (representing the gate capacitances of the neuro-transistors). Figure R1b shows the original synaptic array weights (devices size $\sim 10\mu\text{m} \times 10\mu\text{m}$) and the weights after scaling with a factor $k = 100$ (device size $\sim 100\text{nm} \times 100\text{nm}$). Figure R1c,d compare the SPICE simulated voltages across the 4 post-synaptic capacitors, identical in both microscale and nanoscale cases, which verifies the node voltage invariance with scaling all mask patterns. (Note the time-multiplexing scheme for synapse programming will not change the result.)

Figure R1 SPICE simulated scaling impact on function and power of a capacitive crossbar network. **a**, The schematic of the 4×4 network of Fig. 5 of the main text, consisting of 4 pre-synaptic neuron voltage sources and 4 post-synaptic neuron capacitors (or the gate capacitances of the neuro-transistors). **b**, The original synaptic weights (devices size $\sim 10\mu\text{m} \times 10\mu\text{m}$) and the weights of scaled devices (device size $\sim 100\text{nm} \times 100\text{nm}$). **c-d**, The waveform of pre-synaptic neuron voltage sources (first row, all pre-synaptic neurons share the same waveform), the voltage waveforms across the post-synaptic capacitors (second row), and the power of pre-synaptic neuron voltage sources (third row, negative power indicates energy flow into the network while positive power reveals that pre-synaptic neuron voltage sources receive energy from the network). The post-synaptic capacitors of both networks experience identical voltages. The power of the scaled network is $\frac{1}{k^2}$ of that of the original network.

In addition, the integrate-&-fire behavior is retained after scaling. This is because the localized filamentary switching of diffusive memristor depends on the amount of voltage flux it has received. Here we simulate the scaling effect on the integration process of a single volatile pseudo-memcapacitor, as shown in Figure R2a. A generalized diffusive memristor model is used to capture the voltage flux induced ON switching, with the block diagram shown in Figure R2b, governed by the following equations³.

$$y(t) = g(\vec{x}, u, t)u(t)$$

$$\frac{d\vec{x}}{dt} = f(\vec{x}, u, t)$$

For a voltage-controlled memristive system, the signal $u(t)$ and $y(t)$ represent the voltage across and the current through the memristor, respectively. In our model, we assume $f(u) = |u(t)|$ to model the non-polar ON-switching caused by Ag filament growth. Please note the choice of $f(u)$ is arbitrary, which does not introduce difference between the case with and without scaling, due to invariant node voltages. The diffusive memristor resistance switches from R_{OFF} to R_{ON} ($R_{ON} \ll R_{OFF}$) once the flux x exceeds the threshold value ϕ_{th} (i.e. $g(x)$ is a shifted step function). Note this model does not cover the RESET process which is spontaneous under zero electrical bias for the volatile memristors. To prove that the integrate-&-fire behavior is invariant with scaling, we performed simulation with the experimentally calibrated model using two sets of circuit parameters as shown in Figure R2c-d, corresponding to that before and after $100\times$ down scaling of photolithograph mask patterns. Please note that R_{ON} does not scale because the filamentary switching is localized. Figure R2c shows the simulated integrate-&-fire behavior of a microscale pseudo-memcapacitor, consistent with the experimental observation shown in Fig. 1d of the main text. More importantly, it is observed that the integrate-&-fire behavior of a scaled device (with scaled parameters) in Figure R2d is identical with that of the microscale device, implying the integrate-&-fire behavior is scaling invariant.

Figure R2 Simulated scaling impact on the integrate-&-fire of a pseudo-memcapacitor. a, Circuit of the volatile pseudo-memcapacitor scaling simulation. b, Equivalent block diagram of the simple diffusive memristor model. The absolute voltage across the diffusive memristor is integrated over time, representing the voltage flux which relates to the amount of Ag within the dielectrics. The device resistance switches from R_{OFF} to R_{ON} ($R_{ON} \gg R_{OFF}$) once the flux exceeds the threshold value $\phi_{th} = 3.75 \times 10^{-5} V \cdot s$. c-d, Simulated membrane potential responses before and after $100\times$ down scaling of photolithograph mask patterns, respectively. The responses are identical. Note that $R_{ON} = 1k\Omega$ is not scaled due to the filament nature.

We also experimentally fabricated nanoscale volatile pseudo-memcapacitor as shown in Figure R3a,b. Like the microscale device of Fig. 1a of the main text, it consists of 3 electrodes. The lower two electrodes embed the series capacitance while the upper two connects to the integrated diffusive memristor with $\sim 100nm \times 100nm$ junction size. The nanoscale diffusive memristor shows consistent electrical responses under voltage sweeps with that of microscale devices, which verifies that the Ag dynamics oriented switching of diffusive memristor is localized and scaling independent. (Please note that the inevitable capacitance of the oscilloscope probes in our testing system prevents electrical measurement of the “membrane potential” in the phase of integration within the nanoscale memcapacitors.)

Figure R3 Nanoscale diffusive memristor based volatile memcapacitor. **a-b**, SEM images of the nanoscale memcapacitor. The bottom electrode (BE) and middle electrode (ME) embed the series capacitance, while the top electrode (TE) and ME connect to the integrated diffusive memristor with $\sim 100\text{nm} \times 100\text{nm}$ junction size. **c**, The repeatable bipolar threshold switching of the individual Pt/Ag/SiO_x/Ag/Ag/Pt diffusive memristor.

(2) **Scaling impact on operating power.** The capacitive crossbar network features low operating power as pseudo-memcapacitors store electric energy in electrostatic field. Scaling all lithography mask patterns will further lower down the overall operating power because of two reasons. The first is that the scaled capacitive elements are of less energy storage capacity, thus requiring less driving power of pre-synaptic neuron voltage sources. On the other hand, the neuro-transistors, after scaling, could feature a smaller Joule heat loss due to the increased resistance of pull-up resistors. For the electrostatic fields, the energy stored in a MIM capacitive element is given by the equation below:

$$E_{after} = \frac{1}{2} C_{after} V^2 = \frac{1}{2k^2} C_{before} V^2 = \frac{1}{k^2} E_{before}$$

The stored energy after and before scaling are denoted as E_{after} and E_{before} , respectively. The voltage across the capacitive element is V . Figure R1c,d compares the simulated power of network before and after scaling. The original network is of $\sim 15\mu\text{W}$ peak power in contrast to the $\sim 1.5\text{nW}$ peak power of the scaled network. It shall be noted that ramping up the pre-synaptic neuron voltages (e.g. from time 0 to $1\mu\text{s}$) charges the capacitive network, which withdraws energy from the signal sources. In contrast, ramping down the pre-synaptic neuron voltages (e.g. from time $3\mu\text{s}$ to $4\mu\text{s}$) discharges the crossbar array. Therefore, the pre-synaptic neurons receive energy from the capacitive network from time $3\mu\text{s}$ to $4\mu\text{s}$. At steady state (e.g. from time $1\mu\text{s}$ to $3\mu\text{s}$), the net power is zero.

For the heat loss of the pull-up resistors (much larger than ON-state transistor channel resistance) of the neuro-transistors (See Fig. 1e of the main text), the ON switching of the volatile memcapacitive gates introduce current flows across the pull-up resistors which convert electricity to Joule heat. The resistance of the pull-up resistors scales according to the following equation.

$$R_{after} = \rho \frac{k^2 d}{A} = k^2 R_{before}$$

The resistance after and before scaling are R_{after} and R_{before} , respectively. The resistivity of the resistive medium is ρ . The scaled power is given by the equation below.

$$P_{after} = \frac{V^2}{R_{after}} = \frac{V^2}{k^2 R_{before}} = \frac{1}{k^2} P_{before}$$

The equation implies that the heating power P_{after} scales with the same factor as the electrical charging/discharging power. It is also much smaller than the original power P_{before} . In addition, it shall be noted that the simultaneous scaling of both capacitive and resistive elements will not affect the RC time constant for charging the next stage, which makes the response of network invariant.

Part of the discussion above has been incorporated into the new Supplementary Note 1. Figure R2 and Figure R3 are with the new Figure S11 and S12.

Reviewer #2 (Remarks to the Author):

The manuscript by Wang et al. reports very interesting results on combining the functionalities of two different type devices based on Ag:SiO₂ and Ta₂O₅ implementing them in a neuro-transistor. The authors demonstrate a Hebbian-type (associative) learning mechanism.

The experiments are carefully performed and the manuscript is well structured and written. The topic is suitable for the journal.

I would recommend the manuscript for publication, provided the authors address some issues on the interpretation of the mechanism of operation.

Reply: We thank the referee for the positive comments.

Comments:

1. The authors suggested that Pt/Ta₂O₅/TaO_x/Pt cell is a capacitor. However, several groups have reported that device with this structure works as a memristor (e.g. Nat Commun 2013, doi:10.1038/ncomms3382). Could the authors provide an experimental proof that the cell is indeed memcapacitive and not memristive? This point is important to be justified as it influences the whole discussion and is also present in the title.

Reply: We thank the referee for this comment. We agree with the referee that the Pt/Ta₂O₅/TaO_x/Pt cell could work as a memristor as discussed by the mentioned reference (now cited as Ref. 57 in the main text). However, the pre-requisite of resistive switching in the Ta₂O₅ cell is that the device has to be electroformed and subjected to voltages larger than its switching threshold.

For the integrated memcapacitor with a Pt/Ta₂O₅/TaO_x/Pt cell stacked below a diffusive memristor, the Pt/Ta₂O₅/TaO_x/Pt cell is not electroformed. Therefore, it works as a capacitor (instead of a memcapacitor) with a breakdown voltage ~5V (see Figure R4a). Before breakdown, the leakage current increases exponentially with the applied voltage, showing the typical field dependence of amorphous insulator⁴. The magnitude of the DC leakage current is ~10⁻¹¹A at 2V bias, as shown in Figure R4a. Together with the diffusive memristor integrated with this capacitor, a system with signal history dependent capacitance is formed in this study, also termed dynamic pseudo-memcapacitor (DPM) in

the main text. In addition, if the Pt/Ta₂O₅/TaO_x/Pt cell is electroformed, it exhibits the typical bipolar resistive switching behaviors of oxide memristors. (see Figure R4b)

Figure R4 Forming and typical switching of the Pt/Ta₂O₅/TaO_x/Pt cell of an integrated pseudo-memcapacitor. **a**, Forming I-V and the followed RESET of the first time. It shall be noted that the DC leakage current is $\sim 10^{-11}$ A at 2V bias. **b**, Repeatabile bipolar resistive switching of the cell after forming.

We also compare the impedance spectrum of the pristine Pt/Ta₂O₅/TaO_x/Pt cell, which is capacitor like, with that of the cell after electroforming. (See Figure R5) The measured capacitance component of the impedance is ~ 10 pF as shown in Figure R5a, consistent with the cell geometry in Fig. 1a of the main text. The frequency dependent conductance follows the typical power law for amorphous insulators.⁴ On the contrary, the cell after forming features a frequency independent conductance ~ 14 mS which is likely due to the metallic filament as revealed in Figure R5b. Please note large conductance reduces the measurement accuracy of the associated capacitance component.

Figure R5 Impedance of Pt/Ta₂O₅/TaO_x/Pt cell before and after forming. **a-b**, Measurement schematics and impedance spectrums of pristine and formed Pt/Ta₂O₅/TaO_x/Pt cells, respectively. The measured capacitance is ~ 10 pF of the pristine cell with a conductance component following the typical power law for amorphous insulators.⁴ The cell after forming features ~ 14 mS frequency independent conductance.

2. The cell is very similar to CRS cells, employing one ECM and one VCM cell. Is it possible that this combination is providing the functionalities and not a memcapacitive effect?

Reply: We thanks the referee for raising this possibility. We have performed experiments to exclude the scenario that the observed capacitive switching effect originates from complementary resistive switching (CRS).

We built a volatile pseudo-memcapacitor using a fixed value (1nF) off-the-shelf ceramic capacitor in series with the diffusive memristor. (See Figure R6a or the Fig. S1) The observed charge-voltage response, as shown in Figure R6b, of the wired pseudo-memcapacitor clearly resembles that of the integrated cell in Fig. 1b of the main text. Figure R6c-d illustrate the temporal current-voltage responses with sinusoidal input waveforms of the low capacitance state and high capacitance state, respectively. The amplitude of the sinusoidal current observed is $\sim 3\text{nA}$ ($\sim 5\mu\text{A}$) with a ~ 90 degrees phase difference from the input voltage waveforms, indicating $\sim 2.5\text{pF}$ (1nF) capacitance. Since the electrical performance of wired pseudo-memcapacitor duplicates the observed capacitance switching of the integrated structure, it thus indicates the capacitive switching is not due to CRS.

Figure R6 Memcapacitor built with a wired off-the-shelf series capacitor. **a**, Schematic of the experimental setup for a volatile pseudo-memcapacitor built by wiring a diffusive memristor with an off-the-shelf ceramic capacitor (1nF). **b**, Charge-voltage response of the wired dynamic pseudo-memcapacitor. **c-d**, Equivalent circuits and temporal current-voltage responses of the low capacitance state and high capacitance state of the volatile pseudo-memcapacitor. The amplitude of the sinusoidal current observed is $\sim 3\text{nA}$ ($\sim 5\mu\text{A}$) with a ~ 90 degrees phase difference from the input voltage waveforms, indicating $\sim 2.5\text{pF}$ (1nF) capacitance.

In addition, we also characterized the electrical performance of the integrated pseudo-memcapacitor with a **formed** and switchable Pt/Ta₂O₅/TaO_x/Pt cell in Figure R7. Figure R7a-b show the DC voltage sweeps of both the diffusive memristor and the Pt/Ta₂O₅/TaO_x/Pt cell of a single integrated pseudo-memcapacitor by probing the middle electrode, respectively. Figure R7c shows the voltage sweep response of the whole integrated structure that the amplitude of the current passing through the combined structure is beyond 100 μA once the diffusive memristor is switched ON. This

observation clearly contrasts to that of Fig. S1d-e where the fast voltage sweeping (with a rate $1.6 \times 10^4 \text{V/s}$, $\sim 10^4$ times of the DC voltage sweep rate) is associated with a charging/discharging current of an amplitude $< 250 \text{nA}$. Therefore, the observed capacitive switching is not the same as the CRS like switching in Figure R7, which also indicates that this material stack is capable of two different operation schemes depends on whether the Pt/Ta₂O₅/TaO_x/Pt is formed or not.

Figure R7 Electrical performance of the microscale integrated pseudo-memcapacitor with a diffusive memristor and a **formed** and switchable Pt/Ta₂O₅/TaO_x/Pt cell. **a**, Repeatably bipolar threshold switching I-V characteristics of the individual Pt/Ag/SiO_x:Ag/Ag/Pt diffusive memristor. **b**, Repeatably bipolar memristive switching I-V characteristics of the individual Pt/Ta₂O₅/TaO_x/Pt cell after forming. **c**, DC voltage sweep I-V characteristics of the vertically integrated diffusive memristor and Pt/Ta₂O₅/TaO_x/Pt cell. The diffusive memristor turns on at $\sim 0.5 \text{V}$ followed by the switching of the Pt/Ta₂O₅/TaO_x/Pt cell. It shall be noted that the magnitude of the current flow is larger than $100 \mu\text{A}$ after the ON-switching of the diffusive memristor.

The following sentence is added to the main text. “The Pt/Ag/SiO_x:Ag/Ag/Pt diffusive memristor sits on top of a Pt/Ta₂O₅/TaO_x/Pt capacitor, which could also work as a one-selector-one-memristor cell once the bottom capacitor is electroformed as a nonvolatile memristor at high voltage⁵, making this structure of multiple uses.”

3. The authors are discussing in several paragraphs in the text that their circuit is bio-similar. However, in the living organisms the transmission of the signals occurs based on generated diffusion potential (being a part of the nanobattery effect). To my best knowledge there is no reported biological cell using (bio)capacitors. From the text it can be made a misleading conclusion that implementing capacitors makes the circuit more bio-similar.

Reply: We agree with the referee that the bio-neurons operates on electrochemical effects, similar to nanobattery effects^{6,7} observed in solid-state devices as discussed in the main text. Therefore, we have revised our claim. Instead of “implementing capacitors makes the circuits more bio-similar”, the “pseudo-memcapacitive neuro-transistor provides a better emulation of the biological neural functionalities compared to existing memristive neurons” due to the following two observations, which are detailed in the coming discussions.

- (1) More faithful membrane potential emulation than memristive neurons.
- (2) Active neuron entity with signal amplification capability.

(1) As mentioned by the referee, the temporal summation is powered by the electrochemical gradients (like the nanobattery effects in solid-state devices^{6,7}), which has been mathematically modelled by the Hodgkin-Huxley model⁸. The summation of

signal in time is associated to the switching of the voltage gated sodium and potassium ion channels, which integrates the postsynaptic potentials and initiate the subsequent action potential. As shown in Figure R8a, in a typical temporal summation process, high frequency presynaptic spikes propagate to the soma, which leads to the swift opening and shutting of a small portion of the sodium ion channels and the gradual stepping up of the membrane potential at time t_1 . Once the membrane potential exceeds the threshold at time t_2 , the fast inward-flow of sodium ions results in significant further rise of the membrane potential. This positive feedback rises the potential explosively until all available sodium ion channels are open, leading to the observed large upswing of the membrane potential in Figure R8a. Once reaching the maxima, the membrane experiences repolarization at time t_3 due to the inactivation of the sodium ion channels and the opening of the potassium ion channels.

By virtue of the biomimetic Ag dynamics, the membrane potential to an input pulse train was replicated in the volatile pseudo-memcapacitor. As shown in Figure R8b, the capacitive switch accumulated charge without “firing” due to non-linear OFF state resistance of the diffusive memristor at time t_1' . Such non-linear I-V relation, following typical exponential function, makes the resistance small at large voltage (easy to charge capacitor) but large at small voltage (hard to leak charge), mimicking the function of sodium ion channels. As the charge over the series capacitor increased, the voltage across the series capacitor rose with each subsequent pulse, closely reflecting the expected behavior illustrated for a neuron at moment t_1 of Figure R8a. At time t_2' , the diffusive memristor was switched ON by the pre-synaptic spike to fully charge the capacitor, which replicated the upswing of membrane potential due to opening of all sodium ion channels at t_2 of Figure R8a. At time t_3' , the pre-synaptic input was low. The memristor was first switched OFF and then switched ON again with opposite bias (See Supplementary Information Fig. S2 for the biasing dependent relaxation of the diffusive memristor.), which quickly discharged the capacitor and brought the membrane potential back to its resting value, similar to the repolarization caused by opening of potassium ion channels. The next pre-synaptic input spike would switch OFF the diffusive memristor, and start a new cycle of integrate-and-fire process. The volatility of the filament essentially equips the neuron with the repolarization and self-inactivation features of ion channels, which differs from non-volatile memristive neurons demanding RESET pulses⁹.

Figure R8 Comparison of the integration process of a biological neuron and that of the volatile pseudo-memcapacitor. **a**, Schematic representation of a biological neuron generating an action potential after receiving high frequency postsynaptic inputs. At time t_1 , the membrane potential did not reach the threshold so few sodium channels were open upon the arrival of the signal. As the neuron kept on receiving input stimulus, the membrane potential hit the threshold at time t_2 inducing quick opening of all available sodium ion channels. The potential reached its maxima and started to decrease due to the repolarization caused by the opening of potassium ion channel and inactivation of sodium ion channels. **b**, The integrate-and-fire process of a volatile pseudo-memcapacitor which shares strong resemblance with that of **a**. At time t_1' , the potential across the capacitor rose upon the input stimulus due to the nonlinear OFF state resistance of the diffusive memristor. The resistance was small at high voltage (easy to charge) but large at low voltage (difficult to leak charge), mimicking the sodium ion channel. At time t_2' , the diffusive memristor was switched ON by the pre-synaptic spike to fully charge the capacitor, which replicated the upswing of membrane potential due to opening of all sodium ion channels at t_2 of **a**. At time t_3' , the pre-synaptic input was low. The memristor was first switched OFF and then switched ON again with opposite bias, which quickly discharged the capacitor and brought the membrane potential back to its resting value, similar to the repolarization caused by opening of potassium ion channels.

On the contrary, the recently reported memristor based neuron only showed relatively crude emulation of the integrate-&-fire process. These neurons, according to their constructions and operation mechanisms, could be classified into two categories.

- Category 1: Neurons with dedicated parallel capacitance. This category consists of neuristor¹⁰, Pearson–Anson oscillator with chalcogenide threshold switch¹¹, and diffusive memristor neuron with external capacitance¹². (See Figure R9a-c) For the neurons with dedicated parallel capacitance, the biggest limitation is that the embedded memristors cannot mimic the role of sodium ion channels in the phase of integration. This is because the membrane potential build-up process across the capacitor is independent of the memristor (i.e. ion channel emulator) but solely determined by the configuration of the RC circuit (i.e. the input resistance and the

parallel capacitor), which differs from that of biological neurons. Therefore, the memristor is not capable of capturing the response of sodium ion channels. Instead, the memristor is only used to drain the stored charge once the capacitor potential is beyond its switching threshold, which mimics the function of potassium ion channels. As a consequence, the observed capacitor potential evolution during integration phase shows a clear difference from that of a biological neuron. On the contrary, pseudo-memcapacitor neuron could better emulate both the functions of sodium and potassium ion channels.

- Category 2: Neurons with intrinsic analog switching capabilities. Neurons of this category tend to show intrinsic analog switching capabilities. The conductance of such memristors, when subjected to sub-threshold voltage spikes, tends to gradually increase from insulating value to conducting value. The phase change neuron⁹, Mott insulator neuron¹³, and single diffusive memristor neuron¹² are of this category. (See Figure R9d-f) The neurons based on intrinsic analog switching usually do not have direct physical embodiment of the cell membrane, ion channel, and membrane potential. Instead, the membrane potential is usually mapped to the atomic configuration of the switching material. As a result, there is no associated “ion dynamics”. Compared to intrinsic accumulative switching type neurons, pseudo-memcapacitor neurons possesses physical embodiments of the cell membrane and ion channel, which benefits the fidelity of neural function emulation.

Figure R9 Recently reported memristor based artificial neurons, based on parallel capacitance charging in **a-c** and intrinsic analog switching in **d-f**. **a**, Circuit diagram of the lumped neuristor. The channels consist of Mott memristors (M_1 and M_2), each with a characteristic parallel capacitance (C_1 and C_2 , respectively) and are biased with opposite polarity voltage sources. **b**, Circuit diagram of the Pearson–Anson oscillator with chalcogenide threshold switch neuron. **c**, Circuit diagram of the diffusive memristor neuron with parallel capacitance. **d**, Circuit diagram of the phase change neuron. The neuron stores the membrane potential in the phase configuration of a nanoscale phase-change device. **e**, The Mott insulator neuron. **f**, The single diffusive memristor neuron. (Panel **a** reprinted from Ref. ¹⁰, copyright 2013 Macmillan Publishers Limited. Panel **b** reprinted from Ref. ¹¹, copyright 2016 the Royal Society of Chemistry. Panel **c** and **f** reprinted from Ref. ¹², copyright 2018 Macmillan Publishers Limited. Panel **d** reprinted from Ref. ⁹, copyright 2016 Macmillan Publishers Limited. Panel **e** reprinted from Ref. ¹³, copyright 2017 WILEY-VCH Verlag GmbH & Co.)

(2) In addition, the nanobattery-like effects of biological neurons make the neuron an active voltage source because ion pumps convert chemical energy to electrochemical potential, which makes neurons electrically active elements. Neuro-transistors, built on the pseudo-memcapacitor, have the same capability to drive signals over large fan-out

and propagation in multi-layer networks. On the contrast, most memristor based neurons are passive, particularly intrinsic analog switching neurons (category 2 as discussed in the early text) since the entire neuron is made of a single resistive switching element. The rest memristive neurons, although being active, may show other limitations. The Pearson–Anson oscillator neuron with a threshold switch is active. However, it employs an operational amplifier with tens of transistors compared to the single MOSFET design of the neuro-transistor. The neuristor concept is active and compact. However, it demonstrated tonic spiking with a constant current source rather than being a voltage integrator, which is less compatible with the network design using emerging hardware synapses.

Figure R8 is now with the revised Fig. 1. Part of the discussion has been incorporated into the revised main text.

4. It will be of benefit for the readers if it is more clearly explained what is the advantage of the present work/approach compared to other recent papers published in this field (by the authors and other groups). Most of the recent research in the field is claiming on bio-similarity and it is important to compare and explain in more detail.

Reply: We thank the referee for this constructive suggestion. The proposed neuro-transistor and associated pseudo-memcapacitive network features low power consumption, better emulation of membrane potential in integrate-&-fire, and active neuron operation, as detailed below.

(1) The first advantage of the capacitive neural network compared to the resistive neural network is its low power consumption. As pseudo-memcapacitors store energy in electrostatic field rather than converting electricity to heat, the pseudo-memcapacitive neural network features low energy operation compared to networks built on other emerging devices (e.g. memristors) and free of sneak path leakage issue. The overall power consumption of a memcapacitive network consists of electrostatic energy conversion and heating of pull-up resistors. Both parts could be further reduced by scaling the photolithograph patterns. (See detailed discussion on the scaling benefits to power in the response to the first comment of referee 1.)

The electrostatic energy is proportional to the capacitances of the elements. It shall also be noted, charging pseudo-memcapacitor withdraws energy from the signal sources while discharging pseudo-memcapacitors feeds the signal sources with the electrostatic energy, as the network essentially provides temporary energy storage. At steady state where signal sources outputting constant voltages, the power is zero. This clearly contrasts to the memristive neural network that any non-zero signal will lead to Joule heating at memristive elements.

The other power consumption comes from the pull-up resistors. The neuro-transistors consume static power once the gate pseudo-memcapacitors are switched ON. The current flows from pull-up resistors yield Joule heating, with power determined by the resistance. However, this power is usually very limited as the neurons are normally not firing. In addition, the resistance could be easily increased to reduce heating by scaling, which will also maintain a constant RC charging time of the next stage.

- (2) The second advantage is the ability that the neuro-transistor could better emulate the integrate-&-fire process of a biological neuron compared to other memristor based neurons, as detailed in the response to the third comment of referee 2.
- (3) The third advantage is the sustainable signal propagation thanks to active neuron operation realized on neuro-transistors, which could drive signal over large fan-outs and propagation in multi-layer networks. Please refer to the detailed discussion in the response to the third comment of referee 2.

Part of the discussion above has been incorporated into the main text.

Minor comment:

On page 2 the authors wrote “...conductance resulting from the diffusion of nanoscale Ag clusters in a dielectric medium...”

The sentence is not correct. The Ag-clusters are not directly diffusing but dissolve or agglomerate or are displaced. The Ag⁺ ions are diffusing (more precisely – migrating in the high electric field).

Reply: We agree with the referee that the redox reaction and subsequent ion migration under the external field provide main driving force of Ag mass movement. Such electrochemical migration process has been investigated by Yang et al.¹⁴, Tian et al.¹⁵, and reviewed by Valov et al.⁶, where the bipolar electrodes effects are playing the key roles in low mobility medium (e.g. SiO_x) with relatively low surface redox rate. (See Figure R10) This effect has also been discussed in our early works on the study of a single diffusive memristor and its modeling.¹⁶ The corresponding statement in the main text has been revised as “...conductance resulting from the Ag mass migration due to the combined electrochemical^{6,14,15} and diffusion processes¹⁶ in a dielectric medium...”.

Figure R10 Qualitative model showing filament growth dynamics at different ion mobility and redox rate. (Reprinted from Ref. ¹⁴, copyright 2014, Macmillan Publishers Limited.)

References

- 1 Bakshi, V., Mizoguchi, H., Liang, T., Grenville, A. & Benschop, J. P. Special Section Guest Editorial: EUV Lithography for the 3-nm Node and Beyond. *Journal of Micro/Nanolithography, MEMS, and MOEMS* **16**, 041001, (2017).
- 2 Liu, Q. *et al.* Real-time observation on dynamic growth/dissolution of conductive filaments in oxide-electrolyte-based ReRAM. *Adv. Mater.* **24**, 1844-1849, (2012).

- 3 Pershin, Y. V. & Di Ventra, M. Neuromorphic, Digital, and Quantum
Computation With Memory Circuit Elements. *Proc. IEEE* **100**, 2071-2080,
(2012).
- 4 Wang, Z. *et al.* Transport properties of HfO_{2-x}-based resistive-switching
memories. *Phys. Rev. B* **85**, 195322, (2012).
- 5 Park, G.-S. *et al.* In situ observation of filamentary conducting channels in an
asymmetric Ta₂O_{5-x}/TaO_{2-x} bilayer structure. *Nat. Commun.* **4**, 2382, (2013).
- 6 Valov, I. & Lu, W. D. Nanoscale electrochemistry using dielectric thin films as
solid electrolytes. *Nanoscale* **8**, 13828-13837, (2016).
- 7 Valov, I. *et al.* Nanobatteries in redox-based resistive switches require extension
of memristor theory. *Nat. Commun.* **4**, 1771, (2013).
- 8 Magee, J. C. Dendritic integration of excitatory synaptic input. *Nat. Rev. Neurosci.*
1, 181-190, (2000).
- 9 Tuma, T., Pantazi, A., Le Gallo, M., Sebastian, A. & Eleftheriou, E. Stochastic
phase-change neurons. *Nat. Nanotechnol.* **11**, 693-699, (2016).
- 10 Pickett, M. D., Medeiros-Ribeiro, G. & Williams, R. S. A scalable neuristor built
with Mott memristors. *Nat. Mater.* **12**, 114-117, (2013).
- 11 Lim, H. *et al.* Relaxation oscillator-realized artificial electronic neurons, their
responses, and noise. *Nanoscale* **8**, 9629-9640, (2016).
- 12 Wang, Z. *et al.* Fully memristive neural networks for pattern classification with
unsupervised learning. *Nature Electronics* **1**, 137, (2018).
- 13 Stoliar, P. *et al.* A Leaky-Integrate-and-Fire Neuron Analog Realized with a Mott
Insulator. *Adv. Funct. Mater.*, 1604740, (2017).
- 14 Yang, Y. *et al.* Electrochemical dynamics of nanoscale metallic inclusions in
dielectrics. *Nat. Commun.* **5**, 4232, (2014).
- 15 Tian, X. *et al.* Bipolar electrochemical mechanism for mass transfer in nanoionic
resistive memories. *Adv. Mater.* **26**, 3649-3654, (2014).
- 16 Wang, Z. *et al.* Memristors with diffusive dynamics as synaptic emulators for
neuromorphic computing. *Nat. Mater.* **16**, 101-108, (2016).

REVIEWERS' COMMENTS:

Reviewer #1 (Remarks to the Author):

The authors have answered all my concerns satisfactorily. I therefore think the paper is now ready for publication.

Reviewer #2 (Remarks to the Author):

The authors have addressed my comments and amended/revised the manuscript. I recommend it for publication.

There is only one minor point that can be discussed:

In the unformed device there is still possible variation of the physical effects that can cause same behaviour. For example the device can be a classical capacitor (as discussed by the authors), but it can be also a (electro-)chemical capacitor involving redox reactions at the two interfaces or can be a emf (nanobattery effect based on the chemical asymmetry of the electrodes). Distinguishing these effects is surely not a subject of this paper. I would however recommend to mention these options.

Addressing this point is not requiring a further review.

Reviewer #1 (Remarks to the Author):

The authors have answered all my concerns satisfactorily. I therefore think the paper is now ready for publication.

Reply: We thank the referee for the positive comments on the revised manuscript.

Reviewer #2 (Remarks to the Author):

The authors have addressed my comments and amended/revised the manuscript. I recommend it for publication.

There is only one minor point that can be discussed:

In the unformed device there is still possible variation of the physical effects that can cause same behaviour. For example, the device can be a classical capacitor (as discussed by the authors), but it can be also a (electro-)chemical capacitor involving redox reactions at the two interfaces or can be a emf (nanobattery effect based on the chemical asymmetry of the electrodes). Distinguishing these effects is surely not a subject of this paper. I would however recommend to mention these options. Addressing this point is not requiring a further review.

Reply: We thank the referee for the positive comment. In literature, memcapacitive phenomena could be realized via various mechanisms.¹⁻¹¹ Ionic effect provides one of the most important mechanisms.

As suggested by the referee, electrochemical capacitors may show hysteresis in voltage-bias-dependent differential capacitance, which could potentially yield memcapacitance.¹¹

In addition, the metal-insulator-metal stacks, particularly electrochemical metallization cells, may exhibit voltage-bias-history-dependent built-in electromotive force (emf). The emf is a synergy of the Nerst potential, diffusion potential, and the Gibbs-Thomson potential, collectively termed as nanobattery effect^{12,13}.

What's more, voltage-bias induced ion (or dopant) re-distribution in a host lattice (e.g. Ref. 47 and 49 in the early main text) may also yield memcapacitive effect. For instance, the polarization of the ionic solution due to the slow motion of ions in the host nanopore¹ and mobile dopants that can be repositioned by the voltage³ may lead to voltage-bias-history-dependent polarization, thus showing memcapacitive effect.

In the revised manuscript, the following description is included. “*For instance, electrochemical capacitors¹¹, bias-dependent polarization^{1,3}, and nanobattery effect^{12,13} may all have the potential to implement memcapacitive systems.*”

References

- 1 Krems, M., Pershin, Y. V. & Di Ventra, M. Ionic memcapacitive effects in nanopores. *Nano Lett.* **10**, 2674-2678, (2010).
- 2 Martinez-Rincon, J., Di Ventra, M. & Pershin, Y. V. Solid-state memcapacitive system with negative and diverging capacitance. *Phys. Rev. B* **81**, (2010).
- 3 Bratkovski, A. & Williams, R. S. Memcapacitor. WO2010147588A1 (2010).

- 4 Khan, A. K. & Lee, B. H. Monolayer MoS₂ metal insulator transition based memcapacitor modeling with extension to a ternary device. *AIP Advances* **6**, 095022, (2016).
- 5 Park, M., Park, S. & Yoo, K. H. Multilevel Nonvolatile Memristive and Memcapacitive Switching in Stacked Graphene Sheets. *ACS Appl Mater Interfaces* **8**, 14046-14052, (2016).
- 6 Sarma, S., Mothudi, B. M. & Dhlamini, M. S. Observed coexistence of memristive, memcapacitive and meminductive characteristics in polyvinyl alcohol/cadmium sulphide nanocomposites. *Journal of Materials Science: Materials in Electronics* **27**, 4551-4558, (2016).
- 7 Yang, P. *et al.* Memcapacitive characteristics in reactive-metal (Mo, Al)/HfO_x/n-Si structures through migration of oxygen by applied voltage. *Appl. Phys. Lett.* **108**, 052108, (2016).
- 8 You, T. *et al.* An energy-efficient, BiFeO₃-Coated capacitive switch with integrated memory and demodulation functions. *Adv. Electron. Mater.* **2**, 1500352, (2016).
- 9 Ge, N., Strachan, J., Yang, J. & Hu, M. Memcapacitive cross-bar array for determining a dot product. US20170323677A1 (2017).
- 10 Slesazek, S., Wylezich, H. & Mikolajick, T. in *Circuits & Systems (LASCAS), 2017 IEEE 8th Latin American Symposium on.* 1-4 (IEEE).
- 11 Drüscler, M., Huber, B., Passerini, S. & Roling, B. Hysteresis Effects in the Potential-Dependent Double Layer Capacitance of Room Temperature Ionic Liquids at a Polycrystalline Platinum Interface. *J. Phys. Chem. C* **114**, 3614-3617, (2010).
- 12 Valov, I. *et al.* Nanobatteries in redox-based resistive switches require extension of memristor theory. *Nat. Commun.* **4**, 1771, (2013).
- 13 Valov, I. & Lu, W. D. Nanoscale electrochemistry using dielectric thin films as solid electrolytes. *Nanoscale* **8**, 13828-13837, (2016).